# Study Models for Non-Syndromic Hearing Loss

**DOI:** 10.3390/cells14211658

**Published:** 2025-10-23

**Authors:** Valentine Hoyau, Jean-Christophe Leclère, Stéphanie Moisan

**Affiliations:** 1Univ. Brest, Inserm, EFS, UMR 1078, GGB, F-29200 Brest, France; 2Département D’oto-Rhino-Laryngologie et de Chirurgie Cervico-Faciale, CHU Brest, Univ. Brest, EA4685, LIEN, F-29200 Brest, France; 3Laboratoire de Génétique Moléculaire et d’Histocompatibilité, CHU Brest, Univ. Brest, Inserm, EFS, UMR 1078, GGB, F-29200 Brest, France

**Keywords:** hearing loss, cell lines, animals, organoids, genetic, models

## Abstract

Hearing loss is the most common sensory disorder; It is estimated that nearly 2.5 billion people will have some degree of hearing loss by 2050. Although the causes are diverse, a significant proportion of cases have a genetic origin, which is the main focus of the models discussed in this review. Many loci corresponding to deafness genes have already been identified, and approximately 150 genes are responsible for non-syndromic deafness, which is characterized by partial or total hearing loss that is not associated with other signs or symptoms. Although hearing aids and cochlear implants are widely available today, their effectiveness is often limited, especially in noisy environments, prompting the development of advanced therapies for hearing loss. To evaluate new therapies and improve our understanding of hearing physiology, various models, including cellular, animal, and organoid models, are used to study the inner ear. In this review, we present these different models in detail, with their respective strengths and limitations. This analysis will be particularly valuable in helping researchers to identify the most appropriate model for their specific research questions and to justify their choices from an ethical perspective.

## 1. Introduction

Hearing loss is the most common sensory disorder. More than 5% of the world’s population, 430 million people, including 34 million children, suffer from profound hearing loss. Nearly 2.5 billion people are estimated to have some degree of hearing loss by 2050 [1]. Genetic factors account for approximately half of all cases of hearing loss, with non-syndromic forms representing around 70% of inherited cases [2]. Hearing loss is characterized by an increasing difficulty in hearing, interpreting, and understanding sounds. Hearing loss, even if it is mild, moderate, severe, or profound, can quickly put people at a physical, psychological, and social disadvantage. Hearing loss in children is often detected when the child is unresponsive to speech or has a speech or language delay [3]. Deafness in children can significantly impact communication development, education, and quality of life. In Europe, North America, and most other developed countries, a neonatal hearing screening program has been established, with an estimated prevalence of bilateral hearing loss of 1.33 per 1000 births and 3.5 per 1000 adolescents [4]. The most prevalent type of genetic hearing loss is non-syndromic hearing loss, representing around 70% of hereditary hearing impairment cases. This hearing loss appears without other related physical or developmental issues.

Many loci corresponding to deafness genes have already been identified and mapped [5], and more than 150 genes associated with non-syndromic hearing loss have now been identified [6]. These genes encode elements of the central auditory circuits, the auditory nerve, and the cochlea, and alterations to them can lead to mild to profound hearing loss [7,8,9].

Non-syndromic hearing loss can be autosomal recessive, autosomal dominant, or rarely X-linked or mitochondrial-linked. More than 125 genes are implicated in non-syndromic hearing loss. The loci responsible for non-syndromic hearing loss are called DFN loci for DeaFNess, then ‘A’ denoting autosomal dominant inheritance, ‘B’ autosomal recessive inheritance, and ‘X’ X-linked inheritance [10].

Although hearing aids and cochlear implants are widely available today, their effectiveness is often limited, especially in noisy environments, prompting the development of advanced therapies for hearing loss. To evaluate new therapies and deepen our understanding of hearing physiology, various models are used to study the inner ear, including cellular, animal, and organoid models. This review presents these models in detail, highlighting their respective strengths and limitations, which is crucial for advancing research in this area. We have organized this review by presenting the different study models in four groups: cell lines, animal models, stem cells, and organoids. Based on the literature, we detail the models developed in each group and provide examples of their applications in basic and clinical research.

## 2. Cell Lines

A cell line is a permanently established cell culture that can proliferate indefinitely. Cell lines differ from cell strains in that they are immortalized [11]. They also provide a pure population of cells that allows reproducible results [12]. Cell lines are often easiest to generate from cancer cells. Cancer cell lines often grow without adhering to a surface and can proliferate to very high densities in a culture dish. Similar properties can be experimentally induced in normal cells by transformation with a tumor-inducing virus or chemical [13].

Many cells in primary culture fail to proliferate due to genetic or phenotypic aberrations, terminal differentiation, or a lack of nutrients. In contrast, tumor-derived primary cultures can be cultured for longer periods of time, providing more opportunities to study specific parameters such as genomic alterations, changes in gene expression, or metabolic pathways. The evolution of these cells toward a tumor phenotype leads to genetic instability and selective adaptation of the cells to the culture conditions [14]. It is therefore necessary to carry out studies on immortalized cells. While these cells can arise spontaneously from certain tissues in culture, in most cases, it is necessary to transform them via immortalization genes [15]. Transfection methods require large numbers of cells or depend on the division of target cells. Viral transfer methods, on the other hand, require fewer cells but still involve the division of target cells to integrate immortalized genes into the genome [16]. One way to avoid this problem is to use conditional immortalization genes, which allow the generation of continuously dividing cell lines that can differentiate after inactivation of the immortalization gene.

As hair cells and their supporting cells do not divide, these methods cannot be used. Therefore, cell lines were generated using cells from transgenic H2khtsA58 mice, also known as immortomice [17]. These immortomice are transgenic mice and carry a temperature-sensitive mutant of simian virus 40, which encodes a large thermolabile tumor antigen that can only be immortalized at permissive temperatures [17,18]. As most inner ear sensory cells undergo terminal mitosis early in development and have little regenerative potential, this is of interest [19,20]. The expression of the conditional immortalization gene is under the control of the H2kb promoter, which is active in most cell types and can be upregulated in the presence of interferon γ. In theory, all the cells of the transgenic mouse have a single copy of the gene that is only activated when the correct temperature and interferon are present [21]. When these cells are isolated and cultured at 33 °C in the presence of interferon, proliferation is maintained. However, when the temperature is increased to 39 °C, proliferation stops, and the cells begin to differentiate [22]. Most existing cell lines used to study the inner ear are derived from immortomice.

### 2.1. Ear Cell Progenitor Cell Lines

Ear cell progenitors are being studied with the generation of IMO cell lines. This conditionally immortalized otocyst cell line, derived from 9.5-day-old immortomouse embryos, expresses some genes present in the developing inner ear, such as *Pax-2*, and *engrailed* [23], and in the mature inner ear, such as *myosin VIIa* or *jagged-2*. IMO cell lines can develop characteristics of several mature inner ear cell types in culture. However, the relationship between phenotype development and the differentiation status of IMO cells is not straightforward. Indeed, IMO cells do not phenotypically resemble differentiated cells. Stereocilia will never appear on these cells unless they are transfected with specific cytoskeletal elements. These cells can therefore be used as a model to study early indicators and influences multipotent precursor cells that can direct them toward different differentiated cell pathways [24]. However, these cell lines are numerous and are not commercialized.

### 2.2. Cochlear Cell Lines

The mammalian cochleae contain only a few thousand hair cells, and no regeneration is observed after embryonic development [25,26]. It was therefore necessary to create a cell line derived from the cochlea. The UB/OC-1 and UB/OC-2 cell lines (University of Bristol/Organ of Corti) were derived from immortomouse embryo organs of Corti to model the early stages of hair cell differentiation. These sensory cell precursors were immortalized before full differentiation. Sensory epithelial cells were selected at stage E13 [27]. These cell lines can be grown as proliferating cultures (33 °C, with γ-interferon) or differentiating cultures (39 °C without γ-interferon), due to the temperature sensitivity of the immortalizing gene. These cell lines express hair cell markers, including the transcription factor Brn3.1, the a9 subunit of the nicotinic acetylcholine receptor, and the actin-based motors myosin VI and myosin VIIa. Cytokeratin is normally expressed in all cochlear epithelial cells during embryonic development, but it is downregulated in hair cells. At 33 °C, 10% of the UB/OC-1 cells were labeled with pancytokeratin antibodies, but none were labeled after 14 days at 39 °C. 92% of the UB/OC-2 cells were labeled at 33 °C and 80% were labeled at 39 °C. UB/OC-1 presented relatively low expression of Brn3.1, a9AChR, and myosin VIIa compared to UB/OC-2 at 33 °C, suggesting that UB/OC-1 may have been immortalized at an earlier stage of differentiation. As UB/OC-2 cells express Brn3.1 at 33 °C, this may suggest that they are committed to becoming hair cells. As these cell lines were derived from dissociated cultures, the exact origin of UB/OC-1 is unknown. The genetic profile suggests that UB/OC-1 cells appear to be derived from the greater epithelial ridge, a transient cell population that disappears during postnatal cochlear maturation and has the potential to differentiate into hair cells. Under differentiating conditions, these cells adopt a hair-cell-like phenotype [28]. None of the explored supporting cell markers were expressed, and the cells appeared to commit uniformly to the same phenotype, as shown by the homogeneous expression of brn3c. The proneural gene *Math1*, which is normally expressed in differentiating hair cells at E12.5-E1, is absent in UB-OC-1. Connexins 30, 26, and 43, as well as *α*-*tectorins,* were not observed in UB-OC-1, whereas *β*-*tectorins* were detected at 33 °C and quickly downregulated during differentiation. *Gata3* and *p27^Kip1^*, which are important in inner ear development, are expressed in UB/OC-1 and correlate with the observations made *in vivo*. This could be useful for identifying and dissecting components of signaling cascades that normally have very short and transient expression *in vivo* [29]. The UB/OC-1 cochlear cell line conditionally expresses genes that are essential for normal differentiation. It can be used to identify and characterize genes involved in development, and to screen new therapeutic approaches. UB/OC-1 cells have been used to identify new genetic elements that enable the regeneration of inner ear sensory hair cells. The most prominent microRNAs (miRNAs) in these cells were identified during differentiation towards a hair cell-like phenotype. *In vitro* miR-210 silencing has been performed in UB/OC-1, resulting in hair cell marker expression [30]. miRNA-210 may be a new factor in hearing loss therapy. Identifying the inner ear pathways that it regulates could lead to the discovery of new drug targets for treating hearing loss.

Other cell lines derived from the organ of Corti have been established, such as the House Ear Institute–Organ of Corti 1(HEI-OC1), which is also derived from immortomouse cochleae. The cochleae were taken from 7-day-old mice, and the cells were cultured at 33 °C. The cells were then incubated at 39 °C for different periods of time and allowed to differentiate for 180 days. HEI-OC1 cells cultured under permissive or non-permissive conditions express specific markers for sensory hair cells [22]. These cells are positive for math-1, a transcription factor that is essential for the differentiation of sensory hair cells in the inner ear. They also express prestin, which is a voltage-sensitive motor protein that is responsible for the electromotile response of cochlear outer hair cells. In mature outer hair cells, prestin is exclusively localized to the lateral plasma membrane in the region between the cell nucleus and a zone just below the level of the cuticular plate. In contrast, in HEI-OC1 cells, prestin is localized to the cytoplasm. HEI-OC1 cells also express markers of cochlear supporting cells, such as OCP2 and connexin 26, but only when cultured under permissive conditions. When HEI-OC1 cells are transferred to non-permissive conditions, nestin and connexin 26 are downregulated, and calsequestrin is upregulated. Consequently, there is no evidence to classify HEI-OC1 cells as hair cells or supporting cell precursors. The suitability of HEI-OC1 cells for studying prestin has been investigated using flow cytometry and confocal laser scanning microscopy techniques to describe the pattern of prestin expression in cells cultured at permissive and non-permissive conditions. It showed that prestin translocates from the cytoplasm to the plasma membrane in a time-dependent manner when cells differentiate under non-permissive culture conditions [31]. Further studies on these cells have used CRISPR-Cas9 (clustered regularly interspaced short palindromic repeats) to modify them and ablate Cx43, which is negligible in the mature mammalian organ of corti. This has been done to test the effects of *GJB2* variations, which encode connexin 26 [32]. This enables us to gain a deeper understanding of how these variations induce hearing loss. A full understanding of these mechanisms is essential for establishing a platform for tactical drug design and rational treatment strategies. *GJB2* is the first gene identified that is responsible for autosomal recessive non-syndromic hearing loss and accounts for up to 50% of all cases of pre-lingual hearing loss [33].

### 2.3. Vascular Stria Cell Lines

Hearing loss may be caused by variations in the basal cells of the vascular stria. Therefore, vascular striatum cell lines have been established. The Stria Vascularis K-1 (SVK-1) cell line, derived from the vascular striatum of a 14-day-old immortomouse [34], represents the basal cells of the vascular stria and is an excellent model for studying its function. The SVK-1 clonal line also expresses mRNAs for basement membrane components and basement membrane-associated proteins. The immunostaining profile of the cultured cells closely mimics the *in vivo* situation. Thus, the SVK-1 clonal cell line, which is derived from strial marginal cells, provides a suitable model for future studies on extracellular matrix regulation [35].

Other vascular striae cell lines have been developed, such as the MCPV-8 line, which is derived from primary cultures of gerbil vascular stria marginal cells immortalized with the *HPV-16 E6/E7* viral gene [36]. Stria Vascularis of the cochlea is responsible for positive endocochlear potential and the high K^+^ and low Na^+^ concentration of endolymph. It acts as an important blood-labyrinth barrier, tightly regulating the passage of molecules from the blood into the cochlea. It is therefore vital for hearing function [37]. The MCPV-8 cells preserve most of the morphological and physiological characteristics of marginal cells and maintain the major ionic transport channels and enzymes such as apical Na^+^, K^+^ channels, basolateral Na,K-ATPase, and the Na^+^/Cl^−^/K^+^ cotransporter. These cells have therefore been used to study transepithelial K^+^ and Na^+^ reabsorption by cellular cAMP to make a comparison with the cAMP-mediated K^+^ secretion reported in marginal cells of freshly isolated stria vascularis [38].

Although cell lines are well characterized, have a long lifespan, and proliferate rapidly, their genetic manipulation alters their phenotype, native functions, and responses to stimuli. Serial passages of cell lines can also cause genotypic and phenotypic variation over time, and genetic drift can also cause heterogeneity within cultures at a given moment [12]. Although cell lines may exhibit morphological or molecular features resembling those of particular auditory cell types, they do not fully reproduce the complex physiological characteristics of native cells. Studies have demonstrated that the function and regulation of the retinoic acid receptor α are similar between the MSC-1 cell line and primary rat Sertoli cells. However, MSC-1 cells lack some of the immune privilege properties associated with primary Sertoli cells [39]. Therefore, their use requires careful interpretation and validation against *in vivo* models. Each cell line has its advantages and disadvantages, making them useful for specific applications (Table 1).

## 3. Animal Models

The concept of an animal model used in biological and biomedical research can be broadly defined as “a living organism in which normative biology or behavior can be studied. In addition, a spontaneous or induced pathological process can be investigated, in which the phenomenon is similar in one or more respects to the same phenomenon in humans or other animal species” [40]. In the absence of a satisfactory human cell line with characteristics similar to those of the developing inner ear, primary cultures or animal models can be used to study the interactions between proteins expressed in the inner ear, their spatial and temporal expression patterns, and their functions, among other biological studies.

### 3.1. Mice

Mice can be used to study the development of inner ear defects and to identify the specific functions of genes [41]. The laboratory mouse is widely accepted as an invaluable model organism for studying the genetic basis of human disease, thanks to its short lifespan, ease of experimental manipulation, limited genetic heterogeneity, and controlled handling environment [42]. Mice also offer advantages for analyzing the human auditory system, with its remarkable structural similarity to the human ear. Auditory mouse mutants have provided valuable insights into human ontogeny, morphogenesis, and function. However, the hearing range of mice is about 1 to 100 kHz [43], whereas that of humans is between 20 Hz and 20 kHz. The mouse genome has been fully sequenced and is 80% homologous with the human genome. Mouse genes involved in hearing also exhibit strong sequence similarities and similar functions to their human counterparts [42].

Over the last years, molecular biology techniques and targeted mouse genetics have considerably improved our understanding of the cellular and genetic processes involved in the formation of the cochlear duct and the organ of Corti in the ventral region of the inner ear [44]. Mutant mouse models of inherited hearing loss due to inner ear defects can help identify genes involved in inner ear development or function. Many deafness-related genes were first identified in hearing-impaired mice before being identified in humans. Variations in more than 172 different genes have been reported to be responsible for inner ear malformations or dysfunction in mice [41]. Variations suspected of causing hearing loss can be engineered in mice. Gene-targeted mutagenesis, or knock-out, can be used to reveal a gene’s function by comparing it with that of wild-type mice.

Knock-out and knock-in in mice can be used to demonstrate the role of a novel variant in pathology. To demonstrate the impact of a mutation in *cingulin*, a cytoskeleton-associated protein localized at the apical junctions of epithelial cells, knockout and knock-in mouse models on a *C57BL/6J* background have been performed. These studies revealed that hair cell-specific *cingulin* knock-out leads to high-frequency hearing loss, and *cingulin* variation knock-in mice exhibit noise-sensitive, progressive hearing loss and outer hair cell degeneration. This study reports the identification of the human *Cingulin* gene as a novel cause of autosomal dominant non-syndromic hearing loss. It also provides evidence of the mechanisms by which the variant causes hearing loss, enabling a link to be made between genotype and phenotype [45].

Knockout can also be used to identify new potential therapies. A mouse model of human *DFNA15* deafness, with a *Pou4f3* gene variation, has been engineered. These *Pou4f3*(Δ/+) mice exhibited progressive deafness similar to that observed in *DFNA15* patients. Using *Pou4f3(−/+)* heterozygous knockout mice, it was shown that *DFNA15* is likely caused by haploinsufficiency of the *Pou4f3* gene. Inhibiting retinoic acid signaling using aldehyde dehydrogenase and retinoic acid receptor inhibitors promoted *Pou4f3* expression in the cochlear tissue and suppressed the progression of hearing loss in the mutant mice [46].

However, knockout is not always feasible, so conditional knockout technology may also be used. For example, studies of the *GJB2* gene required a floxed model Gjb2loxP/loxP, which was crossed with mice expressing Cre recombinase under the control of different promoters to drive *GJB2* ablation in a cell-type- or time-specific manner [47]. Indeed, *GJB2* homozygous mutant mice generated by gene targeting died in utero at around day 11 because murine Cx26 plays an essential role in the transplacental transport of glucose, and possibly other nutrients, from the mother’s blood to the fetus [48]. To test variations, a genomically humanized mouse model can also be used, whereby selected mouse genome sequences are swapped with the corresponding human sequence [49]. The CreER-loxP system can also be used to delete transcription factors at different differentiation stages to investigate their role in auditory function, as well as in the survival and maintenance of cells. Using this system, *Pou4f3* has been deleted from mouse cochlear hair cells at different postnatal stages corresponding to specific hair cell maturation and hearing function stages. This has demonstrated that *Pou4f3* is essential for the survival of cochlear hair cells and normal hearing at all postnatal ages [50].

Mice can also be used as models to investigate protein expression in different tissues, providing a better understanding of its implications in pathologies. *TBCD1D24*, which is associated with various phenotypes including non-syndromic hearing loss, has been studied using immunolabelling throughout the postnatal maturation of the cochlear sensory epithelium in the mouse (*BALB/c* strain). Immunolabelling provides precise information on protein localization during development and can be used to investigate its different roles depending on the cell in which it is located [51].

Once the causes of the pathologies had been identified, it was necessary to look for a treatment solution; the most common strategy for inner ear gene therapy is gene replacement. With this approach, a normal copy of the gene’s cDNA is delivered into the inner ear. This therapy is based on a gene delivery vector. The most common approach is the use of the Adeno-associated virus (AAV) vector-based approach. This approach has several advantages, including good efficiency, stable transgene expression, a large cellular tropism, and low immunogenicity. However, it has the disadvantage of a small packaging capacity of only approximately 4.2 kb [52]. This technique is a good option for loss-of-function variants in most autosomal recessive disorders, and for dominant disorders with haploinsufficient loss-of-function variations. The most frequent targets of this strategy are variations in *OTOF*, *GJB2*, *TMC1*, *CLRN1*, and *GJB6* [53]. *OTOF* gene therapy has recently been successfully tested in clinical trials. AAV-OTOF was delivered to one cochlea of a 5-year-old deaf patient and to both cochleae of an 8-year-old deaf patient with *OTOF* variations. This study demonstrates the safety and efficacy of AAV-OTOF in patients [54]. A new study of dual-vector adeno-associated virus gene therapy, which delivers a functional copy of the *OTOF* gene, has recently been reported. Of the 12 participants, who were aged between 10 months and 16 years, 11 experienced a clinically significant improvement in their hearing, and three achieved normal hearing levels. The treatment was well tolerated, with no serious adverse events related to the therapy reported. These results suggest that this therapy could transform the treatment of children with profound genetic hearing loss [55].

To effectively implement these strategies, it is essential to establish a robust study model. Indeed, humans with *GJB2*-related deafness retain at least some auditory hair cells and neurons, and their deafness tends to be stable. In contrast, mice with conditional loss of *Gjb2* in supporting cells exhibit extensive loss of hair cells and neurons and rapidly progress to profound deafness. Therefore, a less severe *GJB2* animal model has been generated with inducible *Sox10iCre^ERT2^*-mediated loss of *GJB2*. In this model, reduced connexin 26 expression and impaired function are observed, but cochlear hair cells and neurons survive for 2 months. *Sox10iCre^ERT2^*; *Gjb2^flox/flox^* mice are valuable for studying the biology of connexin 26 in the cochlea. They may also be useful for evaluating gene therapy vectors and developing therapies for *GJB2*-related deafness [56]. Mouse models are then used to develop these therapies after identifying the gene responsible for deafness. Indeed, after demonstrating *OTOF*’s involvement in *DFNB9* deafness [8] studies have demonstrated that delivering *OTOF* cDNA via AAVs into hair cells is sufficient to restore the hearing of *OTOF ^−/−^* mice. But the *OTOF* coding sequence, which is about 6 kb in size, must be delivered via a dual-AAV strategy. Following the injection of the recombinant vector, via this strategy, into the cochleae of adult *OTOF ^−/−^* mice, otoferlin was detected in the inner hair cells of the entire treated cochlea. This *DFNB9* mouse model demonstrates that cochlear delivery of a fragmented cDNA using a dual-AAV vector can restore production of the full-length protein, even after hearing onset [57,58].

In addition to the dual AAV approach, alternative strategies have been developed to deliver large transgenes, such as single “overload” AAV constructs, or the use of different viral vectors (e.g., lentivirus or adenovirus), which may offer improved cargo capacity or tropism for specific auditory cell types. The “overload” approach of AAV constructs enables the full-length *OTOF* coding sequence to be delivered with a single AAV [59]. Gene replacement is a useful strategy for treating loss-of-function mutations, but it is ineffective for loss caused by mutations that lead to the production of an abnormal protein product, which can interfere with normal cellular function. Other strategies that inhibit the expression of the dominant mutant alleles are required [52]. Gene editing technologies have been developed to correct genetic mutations by adding, deleting, or replacing mutated DNA sequences. The CRISPR-Cas9 system induces targeted DNA double-strand breaks to induce the desired DNA modification. Although CRISPR-Cas9 technology can be used to generate new models, the principal goal is to correct mutations [53]. The *Atp2b2^Obl/+^* mouse, a dominant deafness mouse model carrying a mutation Oblivion (Obl) in the second isoform of the *Atp2b2* gene, has been used to evaluate the delivery of a gene-editing complex targeting the outer hair cells via liposomes. The liposome-mediated *in vivo* delivery of CRISPR-Cas9 ribonucleoprotein complexes leads to Obl allele-specific editing. This *in vivo* genome editing promotes the survival and restores the function of outer hair cells, leading to hearing recovery. It demonstrates the feasibility of ribonucleoprotein delivery of editing agents into the inner ear to target outer hair cell variations [60].

These studies allow us to better understand pathophysiological processes involved in sensorineural hearing impairment, paving the way for the development of inner ear gene therapy. Forty-one proof-of-concept studies for the positive effects of gene therapy approaches in deaf mouse mutants have already been obtained. But successful interventions are restricted to a short time window. Indeed, positive effects have been reported for inner ear interventions when performed during the early neonatal period. So far, only two mouse mutants (adult mice) have been reported to experience hearing restoration: *Vglut3 ^−/−^* or *Slc17a8 ^−/−^* mice and *OTOF ^−/−^* mice [61].

### 3.2. Rats

The rat is a good model for studying hearing because, like the mouse, its cochlear anatomy and physiology are highly similar to those of humans. The large size of their inner ears facilitates *in vivo* manipulation and provides more tissue for ex vivo analysis. Their larger body size also makes them more suitable for repeated screening, which is critical for longitudinal studies [62]. In addition to their anatomical and physiological features, rats are more social than mice and exhibit behaviors that can be compared to those observed in humans. For example, variations in the *FMR1* gene cause social disorders in rats that closely resemble the social behavioral deficits observed in humans with variations in the *FMR1* gene, a similarity that is not observed in mice [63]. Their larger size facilitates surgical and recording procedures, and their advanced learning abilities enable more complex auditory behavioral testing [63]. Rats’ hearing frequency ranges from about 250 Hz to 80 kHz, which is much higher than that found in humans [64]. Rats can be used to study cochlear maturation in terms of hearing sensitivity and other parameters. Unlike humans, they do not have mature hearing at birth, and the development of cochlear function takes longer than in mice, perhaps due to the relative immaturity of the nervous system at birth [65]. Rats have therefore been used to investigate the role of proteins in cochlear development. LaminB1, which plays a key role in development and organogenesis, has been studied in the rat cochlea to determine its role in cochlear development. Its localization and temporal expression pattern in the developing rat cochlea have been investigated using immunofluorescence, Western blot, and quantitative real-time polymerase chain reaction in Sprague-Dawley rats from postnatal day 0 to 21. This study demonstrates LaminB1 as a key regulator for development and differentiation in certain tissue and cell types and provides a theoretical basis for further studies on the physiological function of LaminB1 in the cochleae [66].

Another advantage is the wide variety of strains available; more than 60 species of rats are available [67]. Rats have been used as a disease model for *Pcdh15* deficiency, which is implicated in *USH1F* and *DFNB23*. *Pcdh15* plays an important role in the morphogenesis and cohesion of stereocilia bundles, as well as in maintaining retinal photoreceptor cells. A histological study revealed severe defects in cochlear hair cell stereocilia and the collapse of the organ of Corti. There was also a marked reduction in ganglion cells in adult *Pcdh15^kci^* mutants and a reduction in sensory hair cells in the saccular macula [68].

Although the anatomical structures of the rat cochlea are similar to those in humans and include the presence of the organ of Corti, the tectorial membrane, and Reissner’s membrane, the presence of Deiters’ phalangeal cells and absence of Hensen’s cells make rats less suitable for modeling the functionality of cochlear supporting cells in humans [69].

### 3.3. Rabbits

Rabbits are physiologically similar to humans, widely bred, and very economical compared to the cost of larger animals. Another advantage is their short lifespan. They have acute hearing, with a frequency range of 360 to 42,000 Hz. This is closer to the human range than to the range of mice or rats [70]. They have been used in studies of noise-induced hearing loss [71], presbycusis [72], and audiological research [73]. Recently, the advent of gene editing technology in rabbits has significantly enhanced their value in biomedicine by facilitating the development of more accurate models of human diseases. One such model is the *USH2A* rabbit, which was developed using CRISPR/Cas9 technology and shows that disruption of the *USH2A* gene in rabbits is sufficient to induce hearing loss and progressive photoreceptor degeneration, thereby mimicking the clinical features of *USH2A* disease [74]. This model has been used to better understand the disease with multimodal imaging techniques. This study contributes to our understanding of *USH2A* syndrome and may have potential implications for the development of diagnostic and therapeutic strategies [75].

### 3.4. Zebrafish

The zebrafish is widely recognized as a good model for studying the developmental and genetic features of vertebrates, including the auditory system. Indeed, its genome has been fully sequenced, revealing 71% similarity between zebrafish genes and human genes. The inner ear is accessible, and the tissue of larval fish is transparent [76]. Furthermore, there are similarities between the auditory systems of humans and zebrafish. The zebrafish inner ear contains three semicircular canals, each with a sensory patch, known as a crista, which is composed of both supporting and hair cells. The inner ear also contains two other sensory patches: the maculae and the otolith, which, respectively, detect motion and sound. However, in larval zebrafish, the saccular macula detects sounds, but in adult zebrafish, sound is detected via the saccule and another macula, the lagena, which is a major difference between zebrafish and mammals [77]. While zebrafish possess hair cells that share molecular features with mammalian auditory cells, their overall auditory system is structurally distinct. Zebrafish are therefore most valuable for investigating hair cell development, function, and regeneration rather than for directly modeling mammalian ear morphology. The zebrafish hearing range is about 100 and 5000 Hz [78]. Zebrafish are used to determine the role of proteins and demonstrate the involvement of proteins that are already known to be associated with other pathologies in deafness. The *dhx38 ^−/−^* mutant zebrafish (*dhx38* knockout zebrafish) has been used to elucidate the role of DHX38. This model exhibits significant inner ear impairments, including smaller otocysts, smaller otoliths, and undeveloped semicircular canal projections. These impairments are accompanied by *p53* up-regulation, which leads to apoptosis in inner ear cells. This study suggests that *dhx38* plays a role in the DNA damage response and is required for normal inner ear development by regulating the alternative splicing of several genes involved in this process. Thus, this study provides new insights into the role of *dhx38* in inner ear development [79]. Similarly, a homozygous missense variant in the *oxr1* gene, already known to be associated with neurodegenerative diseases and identified in a 4-year-old girl with sensorineural hearing loss, has been modeled in zebrafish. This gene is expressed in the statoacoustic ganglion and posterior lateral line ganglion in zebrafish, and knockdown of *oxr1b* in zebrafish resulted in significant developmental defects of the statoacoustic ganglion and posterior lateral line ganglion. In addition, it has been demonstrated that this phenotype can be rescued by co-injection of wild-type human *OXR1* mRNA [80]. However, there are some limitations, as the whole genome duplication event occurred in the teleost lineage after it diverged from tetrapods. Even if some gene duplicates were lost over time, it is estimated that 20% of duplicated gene pairs were retained. So, there can be two paralogs for a single mammalian gene [77].

### 3.5. Pigs

Pigs are widely used in biomedical research thanks to their genetics, anatomical, and physiological similarities to humans. They exhibit a hearing range comparable to that of humans and naturally experience age-related hearing loss. The cochleae of pigs and humans are similar in size and structure, and both have a comparable organ of Corti, which comprises hair cells, pillar cells, Deiters’ cells, Hensen’s cells, Bottcher cells, and inner and outer sulcus cells. Furthermore, the human and porcine cochleae are both fully developed at birth and undergo similar developmental trajectories [81]. Various genetically modified porcine models for deafness have been developed. Using CRISPR/Cas9 technology, a porcine model with targeted *OSBPL2* gene deletion has been generated in order to study the molecular pathogenesis of *DFNA67* [82]. Porcine fetal fibroblasts, from Bama miniature pigs, were isolated from the skin of a 1-month-old female Bama miniature fetus and then transfected with a Cas9-sgRNA construct to achieve *OSBPL2* disruption. These donor cells were microinjected into the perivitelline space of the enucleated mature oocytes, which were then transferred surgically into the oviducts of estrus-synchronized pigs. The cloned piglets were subjected to phenotypic characterization of auditory function and serum lipid profiles, showing dual phenotypes of progressive hearing loss and hypercholesterolemia that faithfully resembled the auditory disorder in patients with *OSBPL2* variations. This model contributed to improving our understanding of the pathogenesis of *OSBPL2* deficiency, demonstrating a potential link between auditory dysfunction and dyslipidemia. The Bama Miniature pig has also been used to study congenital single-sided deafness (CSSD), which involves profound sensorineural hearing loss in one ear alongside normal hearing in the other [83]. An inbred porcine strain, with a mating mood of CSSD with CSSD, has been studied to assess the hearing phenotype of the strain and to study the pathological changes to cochlear microstructures. This inbred porcine strain exhibited high and stable prevalence of CSSD, which closely resembled human non-syndromic CSSD disease. Porcine models allow the accurate assessment of the impact of various types of deafness on auditory function and cognitive abilities, establishing more objective standards for the diagnosis and classification of deafness. However, the absence of readily available inbred pig lines results in unstable phenotypes and significant variability in experimental pigs. Compared to other model animals, pigs require the establishment of well-characterized physiological health indicators and are more expensive due to their longer gestation periods, life spans, and more substantial housing costs [81].

Overall, the majority of these animal models have a different hearing range to humans, but they have a structurally similar inner ear and a fully sequenced genome. All models have advantages and disadvantages that must be considered when planning a study (Table 2).

## 4. Stem Cells

Stem cells have self-renewal capabilities and can proliferate and differentiate into a variety of functionally active cells that can serve in various tissues and organs. There are two main categories. The first comprises embryonic stem cells and induced pluripotent stem cells, both of which are pluripotent. These cells can differentiate into all adult body cells. The second category comprises non-embryonic or somatic stem cells, which are more commonly known as “adult” stem cells. Adult stem cells are found in a tissue or organ and can differentiate to yield specialized cell types in that tissue or organ.

Several human-specific biological phenomena, such as brain development and inner ear formation, are complex to study using animal models. Therefore, pluripotent stem cells are increasingly being used as an alternative to animal models to model development and rare diseases [84]. The differentiation potential of stem cells has been exploited to regenerate hair cells and auditory neurons. The three main types of stem cells used in cell regeneration studies are embryonic stem cells (ESCs), tissue-specific stem cells (SCs), and induced pluripotent stem cells (iPSCs). SCs have a limited self-renewal capacity and can only differentiate into cell types within the organ in which they reside. In contrast, ESCs and iPSCs have unlimited self-renewal capacity and can differentiate into any of the three major germ layer cell types [85].

### 4.1. Embryonic Stem Cells

These cells display an unlimited proliferation capacity *in vivo* and *in vitro* and can differentiate into any tissue derived from the three primary germ layers, meaning they are classified as pluripotent cells. Since the establishment of the first mouse embryonic stem cell line in 1981 and the first human embryonic stem cell line in 1998, ESCs have been cultured *in vitro* and induced to become a variety of different cell types [86]. There are significant differences between mice and humans in early embryonic development, especially in terms of extraembryonic structures. This suggests that there may also be significant differences between human embryonic stem cells (hESCs) and mouse embryonic stem cells [87]. The *in vitro* differentiation of hESCs into hair cell-like cells or sensory neuron-like cells is associated with the expression of several well-characterized cell markers for each stage. hESCs can be cultured indefinitely *in vitro* and retain the expression of key epiblast markers, such as the transcription factors OCT4 and SOX2. Numerous induction protocols have been developed to differentiate ESCs into hair cell-like cells and sensory epithelia. Inner ear tissues generated from human and mouse ESCs can be maintained for many weeks under appropriate culture conditions. Patient-specific disease-associated variations can be introduced into the genome of ESCs via CRISPR/Cas9 genome editing technology to study the molecular mechanisms of genetic hearing loss. ESCs are also powerful tools for studying the effects of “genetic correction” in gene therapy [85]. Although working with ESCs represents a significant advancement in disease modeling and offers promising prospects for discovering new therapeutic approaches, the ethical considerations surrounding their use limit their potential for clinical applications [87].

### 4.2. Tissue-Specific Stem Cells

Tissue-specific stem cells are found in fully developed organs and tissues. Classified as multipotent, they retain the ability to self-renew and differentiate into cells from the same tissue or organ in which they are located [88]. Although adult stem cells are likely present in the inner ear, obtaining them without destroying the organ at the intended restoration site is difficult, and therefore, they are unlikely to be useful in human therapies [89]. However, when removed from the context of epithelial organization and supplied with appropriate growth factors and culture conditions, quiescent adult human inner ear tissue is able to proliferate and form [90]. However, these cells can only be harvested during translabyrinthine surgery. It is also possible to use human postmortem autopsy temporal bone to obtain living inner ear cells with progenitor cell properties. However, the success rate of using post-mortem samples to generate spheres is lower.

Some genes can be studied using mesenchymal cells from tissues that are not involved in hearing. Indeed, the role of the *GJB2* gene has been studied in human exfoliated deciduous teeth (SHEDs) homozygous for the c.35del variant. To this end, exfoliated deciduous teeth were collected from children immediately after they fell out naturally. SHED cultures were then compared between patients and control individuals to assess the role of *GJB2* in stem cell differentiation and the relationship between its loss of function and the expression of paralogous genes [91].

While many genes expressed in non-inner-ear cells may also be relevant in hair cells, *in vitro* systems provide a valuable platform to analyze protein function. Techniques such as heterologous expression and the use of hair cell-like cells, as well as functional assays, enable researchers to examine protein localization, interactions, and physiological roles, thereby providing mechanistic insight that complements *in vivo* studies. Studying proteins is essential for understanding cell differentiation. Coupling protein data with gene expression data is crucial for understanding how cell behavior is regulated. Development cannot be fully explained by gene regulation alone; the status of proteins is also important. Simply showing the presence of proteins is not enough; it is also essential to understand the nature of specific signaling pathways, downstream targets, and inhibitory networks, as well as the kinetics of their activation [92]. For example, the role of the mitochondrial fusion protein OPA1 in adult muscle stem cells has been studied using immunofluorescence microscopy, electron microscopy, qPCR, immunoblot, and RNA-seq. This study reveals that OPA1 and mitochondrial structural plasticity act as a physiological rheostat that controls the depth of quiescence and activation potential of muscle-derived stem cells (MuSCs) [93].

### 4.3. Induced Pluripotent Stem Cells

iPSc are adult somatic cells that have been reprogrammed to an immature, pluripotent state. This process involved four transcription factors: octamer-binding transcription factor 4 (Oct4), sex-determining region Y-box 2 (Sox2), kruppel-like factor 4 (KLF4), and c-MYC oncogene [94]. These cells offer new insights into the fundamental molecular mechanisms underlying human development and degenerative diseases at a molecular level. Unlike human embryonic stem cells, iPSCs do not raise ethical concerns [95]. In the case of inner ear sensory cells, pluripotent stem cells have been induced using protocols that manipulate the signaling pathways active during otic development, such as FGF or Wn, to drive differentiation along the otic lineage [96]. This renewable source of tissues and cell types supports the study of otic development, function, and disease in living tissue [97]. The ability to reprogram somatic cells into iPSCs offers the potential to generate pluripotent patient-specific cell lines that can be used to model human disease. Patient-derived iPSCs provide personalized human models for studying genetic inner ear disorders. iPSCs derived from Pendred syndrome patients with biallelic *SLC26A4* point variations have provided new insights into how pendrin protein dysfunction leads to hearing loss [98]. Two iPS cell lines were also generated from peripheral blood mononuclear cells derived from siblings with the homozygous or heterozygous G45E/Y136X variations in the *GJB2* gene [99]. Similarly, a patient-derived iPSC line carrying the ELMOD3 c.512A > G mutation has been generated to study *ELMOD3*, which is implicated in causing autosomal recessive/dominant non-syndromic hearing loss. Transcriptome analysis was conducted on patient-specific iPSCs and healthy sibling-derived iPSCs to compare the two. Then, the patient-derived iPSC line was corrected using the CRISPR/Cas9 genome editing system and compared with the patient-specific iPSCs [100]. However, the noise introduced by the genetic background could potentially obscure small genetic signals of interest in small samples [101]. Although stem cell-derived cells can exhibit many molecular and morphological features resembling inner ear hair cells, they are not fully equivalent. We therefore refer to them as ‘hair cell-like cells’ to reflect their partial similarity while acknowledging the remaining differences in function and maturation.

These different types of stem cells have different advantages and disadvantages, making them useful for different applications (Table 3).

## 5. Organoids

The term organoid refers to cells that grow in a defined three-dimensional environment *in vitro*, forming clusters that self-organize and differentiate into functional cell types, recapitulating the structure and function of an organ *in vivo* [102]. Organoids are useful tools for investigating developmental organogenesis, as well as the processes of adult repair and homeostasis. Their organ-like organization also makes them valuable tools for disease modeling [103]. They can be used as model systems for precision medicine, such as tumor organoids [104]. Organoids offer a promising platform for studying inner ear development and cellular processes *in vitro*. They may eventually reduce the need for animal experimentation, thus fulfilling the 3Rs policy [105]. After all, animals are living beings that deserve respect and should be treated according to the highest ethical standards [106]. The concept of applying humanitarian techniques in animal research is based on the principles of the three Rs: Replacement, Reduction, and Refinement. The 3Rs are currently embedded in legislation governing the use of animals for scientific purposes, such as Directive 2010/63/EU, of the European Parliament and of the Council of 22 September 2010 on the protection of animals used for scientific purposes [107], which is the European Union legislation that protects the animals used in research [108].

3D culture methods mimic the complex tissue architecture and cellular interactions that are characteristic of organs. Organoids can be derived from pluripotent embryonic stem cells, induced pluripotent stem cells, or from adult stem cells. All of these approaches exploit the infinite expansion potential of normal stem cells in culture [109]. Inducing pluripotent stem cells to become otic progenitors requires a cell culture protocol that replicates embryonic developmental stages. This has allowed the generation of various inner ear cell types, including epithelial, neuronal, and glial cells [110]. Some inner ear organoids have been generated from mouse pluripotent stem cells via TGFβ, BMP, FGF, and Wnt signaling. These organoids contain sensory hair cells similar to native hair cells in the mouse inner ear [111]. However, current organoid models do not fully replicate the complexity of the intact auditory system. Systematic analysis and validation of findings, particularly prior to translation into human studies, still necessitate the use of animal models.

### 5.1. Cochlear Organoids from Mouse Cells

Cells in the organ of Corti lose their proliferative and regenerative capacity during postnatal maturation of the auditory system. Unlike progenitors from other epithelial tissues, such as the intestine, LGR5^+^ cochlear progenitors cannot expand effectively and replace lost hair cells. However, they can be induced into a limited proliferative state under specific *in vivo* conditions [112]. Cochlear organoids are inspired by intestinal organoids. Indeed, Lgr5-expressing cells, which are already known and utilized in the gut, can be induced to undergo limited proliferation when stimulated by Wnt within the typically post-mitotic cochlear sensory epithelium [112]. Culturing Lgr5^+^ cochlear progenitor cells on 3D Matrigel scaffolds allows the cells to grow into colonies and differentiate into hair cell-containing organoids via a combination of growth factors and compounds. When dissociated from the cochlear sensory epithelium, Lgr5^+^ supporting cells form organoids and, following treatment with a combination of growth factors and drugs, differentiate into hair cells at high yields.

Culture of cochlear sensory epithelium dissociated cells in Matrigel, growth factor stimulation, and the modulation of chromatin remodeling by the histone deacetylase inhibitor valproic acid increased the percentage of Lgr5-positive cells in 3D culture [113]. Inhibition of the Notch signaling pathway and activation of the Wnt signaling pathway subsequently promoted the differentiation of Lgr5-positive cells into hair cells. Hair cells derived by these methods exhibit apical stereocilia with a luminal orientation and, interestingly, express markers characteristic of either inner hair cells, such as the glutamate transporter vGlut3, or of outer hair cells, such as the motor protein prestin, but not both. However, cochlear organoids can be generated more efficiently by clonal expansion of Lgr5-positive cells when they are derived from neonatal mouse tissue than when they are derived from the adult cochlea.

The factors used to culture inner ear organoids from LGR5^+^ progenitors provide essential but insufficient cues for the long-term expansion of LGR5^+^. Thus, there is no evidence that organoids generated using these protocols recapitulate the morphological, transcriptomic, or functional characteristics of cochlear compartments. To address this issue, several compounds and growth factors were screened to generate cochlear organoids containing hair cells and transmission neurons [114]. The combination of CHIR99021, a glycogen synthase kinase-3 inhibitor that activates Wnt signaling, and lysophosphatidic acid, an essential bioactive phospholipid, resulted in the highest organoid formation capacity. Y-27632, a Rho kinase inhibitor, was shown to promote the survival of dissociated cells, providing a protective effect from days 0 to 4. LY411575 (LY), a γ-secretase inhibitor, can generate numerous MYO7A^+^ cells and induce apoptosis. Bone morphogenetic protein 4, retinoic acid, and LY promote the differentiation of progenitors into hair cells with an F-actin-labeled stereocilia structure. This culture system allows more hair cells to be identified in the organoids and generates cochlear organoids that possess the major cell types of cochlear sensory epithelia. The function of hair cells within organoids was confirmed through a series of morphological and electrophysiological evaluations, and single-cell mRNA sequencing revealed that the differentiation trajectories of cochlear progenitor cells recapitulated the early development of the cochlea. This model can be used for deciphering the mechanisms behind the proliferation of cochlear progenitor cells, the specific differentiation of cochlear hair cells, and might serve as a tool for screening drugs or genetic vectors in the cochlea. Therefore, while organoids can model certain aspects of sensorineural hearing loss at the cellular level, fully recapitulating hair cell–neuron interactions remains a future goal.

Organoids can provide a solution when animal models are not possible. *Grhl2* KO mice die at mid-gestation, and *GRHL2* is associated with progressive, non-syndromic sensorineural deafness, autosomal dominant type 28 in humans. Therefore, cell culture models have been used to better define the *Grhl2* loss-of-function phenotype during ear development. To this end, homozygous deletions in *Grhl2* have previously been induced in mouse embryonic stem cells to differentiate *Grhl2-KO* ESCs along an otic pathway and form three-dimensional inner ear-like organoids [115].

To focus on specific structures, organotypic cultures of the organ of Corti have been generated and used to study the effects of berberine sulfate on damaged auditory cells. Gelatin, sodium alginate, and polyvinyl alcohol were used to fabricate biomimetic scaffolds via 3D printing. The organ of Corti, derived from the inner ear of newborn mice, was seeded onto the scaffold to construct these structures. The resulting structure exhibited good organ of Corti activity and was morphologically identical to the original. This could be an effective model for developing drugs, cell, and gene therapies for sensorineural hearing loss. Ex vivo cultures derived from the neonatal organ of Corti, referred to here as organotypic cultures, allow the investigation of hair cell and neuronal development and function [97]. Unlike organoids, these cultures are obtained from existing tissue and do not reduce the need for animal experimentation, since animals must be sacrificed for each experiment.

### 5.2. Cochlear Organoids from Human Cells

Although information on human cochlear development is limited due to difficulties in accessing samples [116], it is known that hair cells and their supporting cells originate from SOX2^+^ progenitors in a region of the developing cochlear duct, known as the prosensory domain. A cell population expressing EPCAM and CD271 includes almost all hair cell progenitors within the human cochlear prosensory domain. Therefore, a protocol that relies on EPCAM^+^ progenitor cells from the human fetal prosensory domain can be used to generate inner ear organoids. EPCAM^+^ cells must be collected from the human cochlea between weeks 9 and 12 of development and cultured in Matrigel. The EPCAM^+^ organoids are then transferred into a co-culture with EPCAM^-^ cells via semi-permeable inserts that allow medium exchange but prevent direct contact between the two cell types. After coculturing, MYO7A^+^/SOX2^+^/EPCAM^+^ cells, which are indicative of nascent hair-cell-like cells, can be identified in the organoids. This technique provides a novel approach to isolate, expand, and differentiate hair cell-like cells *in vitro* [117].

As the cells were isolated from fetal samples prior to the appearance of hair cells, in vitro differentiation in this system represents a normal developmental process. After several weeks in culture, organoids contain both sensory hair cells and supporting cells [113].

### 5.3. Inner Ear Organoids from Pluripotent Stem Cells

Otic vesicle-like structures containing functionally mature sensory hair cells can be generated from mouse embryonic stem cells [118]. This highlights the binary mechanism underlying the activation of bone morphogenetic protein (BMP) and the inhibition of TGFβ in the induction of nonneural ectoderm *in vitro*. The first step in definitive ectoderm induction involved inducing nonneural ectoderm via serum-free rapid aggregation methods in the presence of Matrigel. The tissue was then induced to differentiate towards a placodal fate by downregulating BMP signaling and stimulating FGF signaling. [119].

This protocol has also been successfully adapted for use with human embryonic stem cells and human induced pluripotent stem cells, by adjusting the timing to align with human fetal development [120]. These findings demonstrate a binary mechanism of BMP activation and TGFβ inhibition for preplacodal induction to trigger the self-directed induction of sensory epithelia. From these, hair cells with the structural and functional properties of native mechanosensitive hair cells in the inner ear spontaneously arise in significant numbers. This approach can be used as a model system for inner ear development, and provides an easily accessible and reproducible means of generating hair cells for *in vitro* disease modeling, drug discovery, and cell therapy experiments [118,120].

These protocols have been adapted to reduce the heterogeneity in the size, morphology, and efficiency of organoid production via a standardized protocol. A new protocol allowed the generation of mouse embryonic stem cell vesicles, which were subsequently mechanically removed from whole aggregates. When embedded in Matrigel, the isolated vesicles expanded into cyst-like structures and autonomously generated sensory epithelia containing hair cells. Although almost all the vesicles within the aggregate could mature into organoids, the efficiency of organoid production depended on the isolation stage of the vesicle and required supplementation with Matrigel [121]. Furthermore, a precise modulation of the SHH and WNT pathways has been reported to confer a ventral otic phenotype to multipotent otic progenitors [122]. This gives rise to hair cells that exhibit the structural, transcriptional, and functional properties of the two types of cochlear hair cells: inner and outer hair cells, which mature *in vitro*. scRNA-seq analysis identified *NR2F1* as a previously unrecognized candidate for the key transcriptional pathway that is essential for cochlear and vestibular diversification. Further studies are required to elucidate the mechanisms underlying the cross-talk between transcriptional pathways and structural development, and to establish a means to control the generation of inner and outer hair cells [122]. While this organoid contains sensory epithelia and connecting neurons and is therefore a useful tool for studying human cochlear development, it lacks all the components of the central nervous system. A model that recapitulates aspects of patterning, regional specificity, and auditory neural circuitry, therefore, needs to be developed.

iPSC lines bearing patient-specific knock-ins and knock-outs using CRISPR/Cas9 have been generated to assess the pathogenicity of candidate variants in the *FGF3* (Fibroblast Growth Factor 3) and *GREB1L* (*GREB1* Like Retinoic Acid Receptor Coactivator) genes. Following exome sequencing for the genetic evaluation of hearing loss associated with cochlear malformations in three probands from unrelated families, deafness variants of uncertain significance were identified in two recognized genes for deafness and in a candidate gene. iPSCs have therefore been differentiated into inner ear organoids, after CRISPR/Cas9 modification. Differences in organoid size, number of luminal spaces, and lower expression of otic vesicle markers have been observed in both knockout and variant-bearing organoids compared to controls [123].

Finally, the iPSC-based organoid model can be cultured and passaged over a long period of time, with a stable genotype that can be frozen and stably cultured after resuscitation. This suggests the possibility of establishing a biological organ bank. However, while the 3D environment of iPSC-derived organoids is more physiologically relevant than 2D cell culture, the relative simplicity of current organoid culture conditions compared to *in vivo* environments means that organoids are not very physiologically close to the native microenvironment [124].

A combination of trajectory-based analysis via RNA velocity, auditory and vestibular enrichment score analysis, and joint projection with *in vivo* developed hair cells revealed separation into auditory and vestibular hair cell-like cells. Single-cell transcriptomic analyses of human pluripotent stem cell organoids support the general notion that organoid hair cell-like cells default to a vestibular-like fate. Without further guidance, hair cell-like cells differentiate randomly into vestibular and auditory phenotypes, although the extrastriolar vestibular pathway appears to be favored. Thus, efforts to modulate different signaling pathways following the otocyst stage *in vitro* could be an effective method to control region-specific differentiation in an organoid model [125].

Current inner ear organoids do not fully recapitulate the morphological, transcriptomic, or functional compartmentalization observed in the native cochlea. Organoids can be used to validate the effects of known variations and therapeutics on human cells. However, more robust protocols and reference atlases are required for the developing inner ear. Readouts must be optimized, and cutting-edge techniques such as co-cultures must be employed to create the most suitable microenvironment possible. These different organoids have different strengths and weaknesses that must be taken into consideration for different studies (Table 4).

## 6. Conclusions

There are many models that can be used to improve our understanding and testing of new therapeutic methods. These models have different advantages and disadvantages, so it is important to select the most suitable one according to its intended use and limitations. Cellular models facilitate fundamental research by enabling the study of the impact of genetic variations [90], gene regulation [126,127], protein function [89], and pharmacological responses. However, animal models are required for the study of protein interactions, spatial and temporal expression, and function. They have improved our understanding of hearing and can serve as good pathological models. Finally, organoids allow the study of development, cell differentiation, and regeneration.

These models are used in studies aimed at understanding pathologies, improving diagnosis, and developing treatments. Mammals have been studied to improve our understanding of hearing function [128], and are now being used to model and understand pathology and to develop new therapeutic strategies that can restore regenerative cochlear capacity. One such strategy involves using animal models in which cochlear hair cells have been eliminated [129]. Indeed, mice and rats are used in the development of gene therapies aimed at restoring hearing [130]. Other animal models also exist, such as cats, gerbils, and guinea pigs, which allow the study of other types of hearing loss. These include noise-induced and drug-induced hearing loss, as well as vulnerability to noise and otitis media. They also enable otological research.

Organoids are currently a promising platform for generating and studying specific inner ear cell types *in vitro*, providing valuable insights into the development and disease mechanisms of the inner ear. However, they do not yet replicate the complex structural and functional organization of the cochlea, including its compartments, multi-tissue integration, and functional synapses. Nevertheless, they may ultimately inform future stem cell–based therapies.

Although gene modification therapies for hearing loss are not yet widely approved or regulated in most countries, several approaches targeting specific genetic defects are under active investigation. For instance, gene replacement strategies for *OTOF* variations have reached the stage of clinical trials, and preclinical studies have demonstrated the potential of gene editing, antisense oligonucleotides, and RNA-based therapies for other genes associated with deafness, such as *GJB2* and *MYO7A*. These studies illustrate how advances in our understanding of the molecular basis of hearing loss can inform future precision therapies. While clinical application remains limited by regulatory and safety considerations, ongoing research provides a framework for the eventual translation of gene-targeted therapies into clinical practice.

## Figures and Tables

**Table 1 cells-14-01658-t001:** Advantages, disadvantages, and utilization of the different cell lines used to study non-syndromic hearing loss.

Model	Disadvantages	Advantages	Utilizations
IMO cells	Do not phenotypically resemble differentiated cells [24].	Express genes present in both developing and mature inner ear [21].	Studying early developmental indicators and factors influencing multipotent precursor cells toward specific differentiation pathways [24].
UB/OC	None of the explored supporting cell markers were expressed [29].	Conditionally express genes essential for normal differentiation [28].	Identifying and dissecting components of signaling cascades with transient *in vivo* expression [29]. Characterizing genes involved in inner ear development and screening potential therapeutic approaches [28].
HEI-OC1	Cannot be definitively classified as hair cells or supporting cell precursors [22].	Express markers of sensory hair cells [22]. Express markers of cochlear supporting cells [22].	Used with CRISPR/Cas9 to study *GJB2* [32]. Studying prestin function [31].
SVK-1		Immunostaining profile closely mimics *in vivo* basal cells [35].	Modeling basal cell function in the stria vascularis. Studying extracellular matrix regulation [35].
MCPV-8		Preserve morphological and physiological characteristics of marginal cells, including key ionic transport channels and enzymes [38].	Studying trans epithelial K^+^ and Na^+^ reabsorption mediated by cellular cAMP [38].

**Table 2 cells-14-01658-t002:** Advantages, disadvantages, and utilizations of the different animal models used to study non-syndromic hearing loss.

Model	Disadvantages	Advantages	Utilizations
Mice	Phenotypes do not always precisely match those of humans [56].	Short lifespan. Ease of experimental manipulation. Limited genetic heterogeneity [42].	Studying inner ear development and gene function [41]. Demonstrating the role of novel variants via knock-out/knock-in models [45]. Investigating protein expressions in different tissues [48].
Rats	Lack Hensen’s cells [69]. No mature hearing at birth [59]	Large inner ears allow *in vivo* manipulation and yield more tissue for ex vivo studies [62]. Presence of key cochlear structures (organ of Corti, tectorial membrane, Reissner’s membrane, Deiters’ phalangeal cells) [69]. Short lifespan	Studying cochlear maturation, including hearing sensitivity. Investigating the role of some proteins, like LaminB1, in cochlear development [60].
Rabbits		Physiologically similar to humans. Widely bred and easy to maintain. Short lifespan [70].	Studies of noise-induced hearing loss [71], presbycusis [72], and general audiological research [73].
Zebrafish	Sound detection differs (via the saccule and another macula). Presence of gene paralogues complicates interpretation [77].	Transparent larval tissue allows direct observation. Accessible inner ear [76].	Studying developmental and genetic aspects of vertebrate hearing [76]. Investigating protein function [79].
Pigs	Expensive. Long gestation period. Absence of readily available inbred pig lines [81]	Strong anatomical and physiological similarities to humans. Naturally develop age-related hearing loss. Cochleae are similar in size and structure to those of humans. Cochlear cell types (hair cells, pillar cells, Deiters’, Hensen’s, Boettcher’s, and sulcus cells) closely match those in humans. Fully developed cochlea at birth, with a developmental trajectory similar to humans [81].	Studying the molecular pathogenesis of genes [82]. Investigating congenital single-sided deafness [83].

**Table 3 cells-14-01658-t003:** Advantages, disadvantages, and applications of stem cells to study non-syndromic hearing loss.

Model	Disadvantages	Advantages	Utilizations
Embryonic stem cells	Ethical considerations limit their potential for clinical applications [87]	Can differentiate into any tissue derived from the three primary germ layers [86].	Differentiating ESCs into hair cells and sensory epithelia using established induction protocols [79]. Introducing disease-associated variations into the genome [79].
Tissue-specific stem cells	Difficult to obtain without damaging the organ of interest. Harvesting possible only during translabyrinthine surgery [84].	Self-renew and differentiate into cell types of their tissue of origin [88].	Studying gene function in mesenchymal cells, including genes from tissues not directly involved in hearing [91].
Induce pluripotent stem cells	Noisy genetic background [101]	Can generate patient-specific pluripotent cell lines. Model human disease [98].	Investigating basic molecular mechanisms of human development. Studying degenerative diseases at the molecular level [95].

**Table 4 cells-14-01658-t004:** Advantages, disadvantages, and utilizations of the different organoids used to study non-syndromic hearing loss.

Model	Disadvantages	Advantages	Utilizations
Cochlear organoids from mouse cells		Hair cells exhibit apical stereocilia and express markers of inner or outer hair cells [113]. Some hair cells form functional synapses with neurons [114]. Cochlear progenitor cell differentiation recapitulates early cochlear development [115].	Modeling sensorineural hearing loss at the cellular level [114]. Drug, cell, and gene therapy development for sensorineural hearing loss [115].
Cochlear organoids from human cells		Novel approach to isolate, expand, and differentiate hair cell-like cells *in vitro* [117].	Studying human cochlear cell differentiation and modeling hair cell-related pathologies.
Inner ear organoids from pluripotent stem cells		Hair cells spontaneously arise with structural and functional properties of native mechanosensitive hair cells [118,120]. Generate both inner and outer hair cells that mature *in vitro* [122].	Model inner ear development. Generate hair cells for *in vitro* disease modeling, drug discovery, and cell therapy [118,120]. iPSC lines with patient-specific knock-ins/knock-outs via CRISPR/Cas9 [123].

## Data Availability

Not applicable.

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
