# Peer review of "Study Models for Non-Syndromic Hearing Loss"

_cells, 2025, doi:10.3390/cells14211658_

Round 1

Reviewer 1 Report

Comments and Suggestions for Authors

Hoyau and colleagues present a wide overview of different models used to study the auditory system, with the aim to provide a guideline to researchers for chosing the correct model for their own study. While generally they showcase different models quite well, I feel that often the article simply lists different examples of applications for a specific model without much benefit to the reader, while in other cases the overview fails to mention specific models, advantages or shortcomings. My main criticism applies to the model organisms chapter, on which I focus since this is my main area of expertise.

General comments:

Quite often, it did not become clear to me why the authors cite a specific paper - in many cases there are previous publications that should rather be cited. Citing a review is fine for a general comment, but for a specific fact the original publication should be cited, not a review article mentioning this fact.

Not sure if this goes against journal style, but I would prefer if citations are at the beginning of a statement. Often, multiple sentences contain statements from a specific publication, which is then only cited at the end of the paragraph. This leaves the reader to wonder where the information before is coming from. Indeed, in some cases I could not find the information I was looking for in the cited paper.

More specific comments:

Abstract, line 10-11 and intro, line 28: "it is estimated that nearly 2.5 billion people will have some degree of hearing loss by 2050." - immediately after you discuss deafness genes. This is a bit misleading, since of the 2.5 billion people only a fraction will have hearing loss due to genetic causes, which are the mail target for most of the models you are discussing in the article.

Line 49: Why cite this article? This has been described much earlier, e.g. Shearer AE, Hildebrand MS, Odell AM, et al. Genetic Hearing Loss Overview. 1999

Chapter "Cell Lines" - it should be made clearer that the cells appear similar to specific cell types but are by no means identical. If the cells express similar proteins it does not mean that they behave as the "original" does.

Line 293: Alternatives to dual AAV strategy should be mentioned, such as overload AAV or different viral vectors

Line 300: This is another case where I can't follow why a certain article is cited - rather than citing Hu et al., 2024, the more appropriate choice would have been Al Moyed et al., 2019, and Akil et al., 2019, who provided the first data on OTOF gene therapy in mice, rather than citing work that modifies some aspects of the ground-breaking work done before.

Line 315 ff: It should alos be discussed that rats are better suited for behavioral experiments than mice.

Line 345-346: I don't understand why the hearing range of rabbits is discussed and not that of mice, who have a much more different hearing range. Also, the hearing range of humans is unaccurately reported as 63 to 23,000 Hz, while it is 20 to 20,000 Hz.

Line 354-455, 365: I don't agree that the zebrafish is an excellent model to study the auditory system, since its system fundamentally different from that of mammals. Especially the mention that they are commonly used to study ear morphology is misleading.

Table 2 and general comments: There are a number more animal models that are commonly used, e.g. guinea pig, Mongolian gerbil, cat, and marmoset monkey, each with their own pros and cons. The comparison in table 2 is not systematic - a number of pros or cons are listed for one model organism, but actually apply for more, where they are not mentioned. Examples: Hearing range being different from human is mentioned for rabbit, but not for zebrafish, mouse, or rat, where it deviates from the human range much more. For pigs, the similarity to human hearing is mentioned, though pig hearing extends further intro ultrasound than rabbit hearing.

Similarly, it is mentioned for zebrafish that the genome has been fully sequenced - this is also true for the other organisms that are named, so it's no a unique advantage.

For rats, it is mentioned that there is a wide variety of strains available - this is also true for mice.

Also, it does not become clear why rabbits are used for "development of more accurate models of human disease".

Some of the utilizations mentioned emerge directly from specific features of the models, that are however not named. For example, "investigating protein expression in different tissues" as a use case for mouse - mice are especially well suited for this because many antibodies are generated againt mouse proteins, which is done because mice are an established genetic model organism. Neither of these two reasons are named.

Why the authors state that zebrafish are used for "investigating ear morphology" escapes me, since zebrafish ears are quite different from all the other named organisms.

Stem cells chapter: It should be made clear that while the cells that are generated in these trials have sometimes great similarity to inner ear cells such as hair cells, at the current state they should still be referred to as "hair cell like cells", since they are still quite different in some properties.

Line 464-470: This paragraph simply states that genes which are expressed in non-inner-ear cells can also play a role in these cells, which is not surprising. I would rather expect some explanation of how protein function can be examined in in vitro systems to better understand the role of the protein in vivo.

Line 507: "A major advantage of organoids is that they can replace animals, thus fulfilling the 3Rs policy." - I don't agree. Of course it would be great if expeiments in organoids could replace animal experiments, but especially for systemic analyses this is currently and in the foreseeable future not possible. If you want to understand how the auditory system works, you need to examine an auditory system. Organoids can hopefully in the future replace some animal experiments, but validation quite often needs to be done in animal experiments, especially before transferring the results into human application. I think that the later sentence "However, the extent to which inner ear organoids mimic the complexity of the normal inner ear remains unclear. " does not do this justice.

Line 564-566: "Thus, these organoids can be used to model sensorineural hearing loss caused by the degeneration of both hair cells and synapses in vitro" - I understand this statement is taken from the cited paper, but this is incorrect. So far, to my knowledge, no working synapses have been demonstrated in organoids. Some function expected from hair cells has been shown in hair cell like cells, some neuronal function has been shown in SGN-like cells, but the direct communication between these cells has not been demonstrated.

Line 567-574: I personally would not refer to this as an organoid, I would refer to this as organotypic culture, since the organ of Corti is already very much present in newborn mice, the cells have simply not finished maturation. Also, this approach does not fulfill the 3R principle as mice need to be killed for every experiment.

Table 4: In addition to the point about synapses mentioned above: Again, why is it mentioned for human organoids that they require several weeks to mature? This is the same for mouse organoids, where it is not mentioned. 

I don't agree that organoids can already be used for "Modeling sensorineural hearing loss caused by degeneration of hair cells and synapses in vitro." I have not seen evidence of this yet.

In my eyes, it is not "Unclear if organoids fully recapitulate morphological, transcriptomic, or functional characteristics of cochlear compartments", they don't. I have not seen evidence of real compartmentalization in organoids.

Also "Complexity of inner ear structures not fully mimicked" - most of the inner ear structures are simply not present in organoids.

Thus, I also don't agree to the final sentence in the conclusions "Organoids are a promising technique for generating functional cochlear structures in vitro to restore hearing, providing crucial information for stem cell use". They are useful for creating cells, not structures, which are either not present at all (compartments, organization of different tissues) or unclear (synapses). I think organoids are a great tool, but whis article promises some things they won't be delivering in the foreseeable future.

Minor points:

Line 162: Talking about GJB2: Since you mentioned connexin26 before - maybe mention that it is encoded by this gene.

Line 255-256: "genomically humanized mouse model can also be used, whereby the mouse genome sequence is swapped with the corresponding human sequence" - please rephrase to make clear that this does not refer to a swap of the entire genome.

Line 280: It should probably be mentioned that OTOF gene therapy has successfully been tested in clinical trials.

Line 296-297: "Restoring the hearing of adult mice is more meaningful, as the inner ear hair cells of newborn mice are fully developed as in humans." - I assume you mean "not fully developed"

Line 325: "studied in rate cochlea" - change to "rat cochlea"

Line 684: Framented sentence: "The use of inner ear models improve diagnosis by identifying genes involved in hearing cis-regulatory elements"

Reviewer 2 Report

Comments and Suggestions for Authors

General comment:

This is a great study addressing the study model of non-syndromic hearing loss. The authors emphasized that hearing loss is the most common sensory disorder; it is estimated that nearly 2.5 billion people will have some degree of hearing loss by 2050. Many loci corresponding to deafness genes have already been identified, and about 150 genes responsible for non-syndromic deafness, partial or total hearing loss not associated with other signs or symptoms. Although hearing aids and cochlear implants are widely available today, their effectiveness is often limited, especially in noisy environments, prompting the development of advanced therapies for hearing loss. To evaluate new therapies and improve the understanding of hearing physiology, various models, including cellular, animal and organoid models, are used to study the inner ear. Therefore, the authors conducted this review to present these different models in detail, with their respective strengths and limitations.

Generally speaking, the current review was well-written and their methodology sounds scientifically. I read with interest. I believed that this manuscript will bring important scientific contributions to the world.

Minor comments:

  1. I surely recognized the fact that this is a narrative review. However, it would be helpful for readers to follow the authors’ rationale if the authors provided a section of method to explore their review process.
  2. As a clinician, I would have strong interest in the application of these cell line/animal model of non-syndromic hearing loss in real world clinical practice. It would be so great if the authors could provide a new paragraph addressing the application of these information in real world clinical practice.
  3. Further, I surely recognized the fact that, currently, there had not been definite governmental agreement for gene modification therapy in the most countries. However, if it be possible in the future, may the authors provide a summary of treatment applications related to those gene defects related to hearing loss?
  4. Although the authors had provided a graphic abstract (I supposed the figure in page 1 line 21 to be graphic abstract), it consisted of insufficient information to catch up the main findings of this review. Please enhance the information in it.
  5. I surely recognized the fact that it was not routine procedure in animal study. However, has this review been registered in any open registry database before they started their study?

Round 2

Reviewer 1 Report

Comments and Suggestions for Authors

I would like to thank the authors for addressing my previous comments. I find that the article has been significantly improved and have only few minor points:

  • Line 306: By now, a number of patients have been treated in several OTOF gene therapy studies (see e.g. Valayannopoulos et al., DB-OTO Gene Therapy for Inherited Deafness. N Engl J Med. 2025. doi: 10.1056/NEJMoa2400521)
  • Line 354 "restauration" should be "restoration"
  • Line 362: I would remove the part reading "and exhibit behavior more similar than seen in humans", but I would leave this to the discretion of the authors
  • Line 552: I would add "For example, " before "The role of the mitochondrial..."
  • Line 657: Please remove "and functional hair-cells- spiral ganglion neurons circuits". As discussed before, these have not been shown to exist in organoids yet.

I would still like to note that I don't agree with the author's notion that rabbits are one of the most important animal models for hearing. Indeed, pubmed searches show significantly more results when searching for keywords such as "hearing", "auditory", or "deafness" in combination with "guinea pig", "cat", or "gerbil" than in combination with "rabbit".

Reviewer 2 Report

Comments and Suggestions for Authors

Many thanks for your detailed response to my comments. The current version looks so great.
